# Uterine leiomyoma is associated with the risk of developing endometriosis: A nationwide cohort study involving 156,195 women

Kent Yu-Hsien Lin[1,2,3‡], Chih-Yi Yang [4,5‡], Alan Lam[6], Cherry Yin-Yi Chang[4,7]*, Wu-Chou Lin[4,7]*

**1** Department of Obstetrics and Gynecology, Women and Children's Health, Royal North Shore Hospital, Sydney, NSW, Australia, **2** Department of Obstetrics and Gynecology, Bankstown-Lidcombe Hospital, Sydney, NSW, Australia, **3** Department of Obstetrics and Gynaecology, Ryde Hospital, Sydney, NSW, Australia, **4** Department of Obstetrics and Gynecology, China Medical University Hospital, Taichung, Taiwan, **5** Department of Public Health, China Medical University, Taichung, Taiwan, **6** Centre for Advanced Reproductive Endosurgery, Sydney, NSW, Australia, **7** School of Chinese Medicine, China Medical University, Taichung, Taiwan

‡ These authors are co-first authors on this work.
* d4754@mailcmuh.org.tw (CYYC); CMUHLinwuchou@hotmail.com (WCL)

**Data Availability Statement:** All relevant data are within the manuscript and its Supporting Information files.

## Abstract

### Objective

Evidence for an association between uterine leiomyoma and increased risk of endometriosis is limited by small sample sizes and short follow-up periods. We assessed this association in a large nationwide sample with 14 years of data.

### Design

Data were sourced from Taiwan's Longitudinal Health Insurance Database 2000 (LHID2000).

### Materials and methods

We identified 31,239 women aged ≥20 years diagnosed with uterine leiomyoma (International Classification of Disease, Ninth Revision, Clinical Modification [ICD-9-CM] code 218) between Jan 1, 2000 and Dec 31, 2012, who were matched with 124,956 controls (1:4) by 5-year age groups and year of diagnosis. Follow-up was from the date of LHID2000 entry to the first occurrence of endometriosis, loss to follow-up, insurance termination, or until December 31, 2013, whichever was earlier.

### Results

In Cox regression analysis, the adjusted hazard ratio (aHR) for endometriosis in women with uterine leiomyoma was 6.44 (95% CI, 6.18, 6.72) compared with controls. The risk of endometriosis was significantly increased in women with uterine leiomyoma and comorbidities of tube-ovarian infection (aHR 2.86; 95% CI, 1.28, 6.36), endometritis (1.14; 1.06, 1.24), infertility (1.26; 1.16, 1.37), or allergic diseases (1.11; 1.05, 1.17). Having both uterine

**Funding:** The funders had no role in study design, data collection and analysis, decision to publish, or preparation of the manuscript.

**Competing interests:** The authors have declared that no competing interests exist.

leiomyoma and endometritis significantly increased the risk of endometriosis (aHR 6.73; 95% CI, 6.07, 7.45) versus having only uterine leiomyoma (6.61; 6.33, 6.91) or endometritis (1.49; 1.31, 1.69). Similarly, having both uterine leiomyoma and infertility significantly increased the risk of endometriosis (aHR 6.95; 95% CI, 6.21, 7.78) versus having only uterine leiomyoma (6.66; 6.38, 6.96) or infertility (1.78; 1.57, 2.02).

## Conclusions

A diagnosis of uterine leiomyoma appears to increase the risk of endometriosis. Patients presenting with uterine fibroids should be encouraged to give informed consent for possible simultaneous surgical treatment of endometriosis.

## Introduction

Of all solid pelvic tumors, uterine leiomyoma is the most common, affecting as many as 80% of women [1, 2]. While many patients are asymptomatic, leiomyoma-related symptoms commonly include abnormal uterine bleeding, pressure-related bladder or bowel symptoms, pelvic pain, or reproductive dysfunction that may require medical or surgical intervention [1–5]. The pathogenesis of uterine leiomyoma is uncertain, but is likely to involve genetic predisposition, steroidal hormones and growth factors [5, 6]. Endometriosis is another common gynecological condition, affecting 10–50% of reproductive-aged women [7–9]. The most common symptoms are dysmenorrhea, chronic pelvic pain, dyspareunia and subfertility [10–12]. These symptoms can have enormous impact on a sufferer's quality of life, with well-documented evidence describing how these symptoms interfere with daily life and activities, adversely affecting general health and psychological well-being, and impacting on overall sexual functioning [13–15]. Moreover, endometriosis-related symptoms impose a substantial economic burden on those afflicted in terms of direct costs of medical care and indirect costs, such as costs for sick leave and work absenteeism [14, 16]. It is well known that the risk of endometriosis is higher in young women with menstrual dysfunction and any obstruction to the outflow of menstrual products [17, 18]. This is thought to be because of the higher volume of refluxed menstrual blood and endometrial tissue fragments, lending support to the theory that endometriosis results from retrograde menstruation [17]. Genetic insights, the possible roles of the environment and the immune system, as well as intrinsic abnormalities in the endometrium of affected women have helped to elucidate the pathogenesis of endometriosis [19]. Endometriosis is an estrogen-dependent disorder. Recent speculations suggest that the estrogen-dependent activity of different ion channels may contribute to the etiology of endometriosis [20]. The levels of estrogen expression and functionality of these ion channels fluctuate between menstrual phases, with increased estrogen enhancing pathogenic aspects of endometriosis, such as the proliferation and *in vivo* implantation of ectopic endometrium [20]. More evidence is needed to determine the strength of this association. Available conventional therapeutic strategies for managing endometriosis (surgery, hormonal treatment, nonsteroidal anti-inflammatories) have limited or intermittent efficacy, as well as unwanted hormonal side effects such as the loss of bone density [21]. Aromatase inhibitors (AIs) have been proposed for endometriosis treatment, with limited clinical data indicating associated reductions in volume and symptoms of endometrioma, but AIs have also been linked to the development of functional cysts and menopausal symptoms, as well as reductions in bone mineral density [22]. Alternative progestins and estroprogestins drug delivery methods (e.g., vaginal rings,

patches, subcutaneous implants, intrauterine drug delivery systems, nanotechnologies) offer potential advantages of fewer adverse events, greater therapeutic efficacy, improved patient compliance and satisfaction [23]. At this point, more clinical experience is needed before any conclusions can be made about the clinical efficacies of these alternative drug delivery methods [23]. Thus, the disadvantages of current pharmacotherapeutic options have induced many women with endometriosis to seek alternative treatment options, such as phytotherapy [21, 24]. Promisingly, *in vitro* and preclinical investigations have indicated that some phytochemicals demonstrate strong phytoestrogenic effects capable of modulating estrogen and inflammatory activity, which could help to reduce endometriosis symptoms, although randomized controlled trial data are lacking [21, 24]. Interestingly, increasing clinical and pathophysiological evidence supports the notion that chronic inflammation is associated with benign gynecological disorders such as uterine leiomyoma and endometriosis [25]. These associations have been comprehensively reviewed [25], although basic and clinical investigations have yet to clearly define the mechanism responsible for the development of endometriosis. Up until now, research has focused on altered immune responses and other regulatory factors that may mediate the histogenesis of endometriosis.

Uterine leiomyoma and endometriosis are hormone-dependent conditions that share many common clinical features including pelvic pain, menstrual abnormalities and subfertility. However, whether these disorders co-exist by chance or because they share common etiological factors is uncertain, as few studies have investigated the association. One research group from Finland has suggested that symptomatic endometriosis appears to coincide with symptomatic uterine fibroids [26]. In one study involving 3,684 Italian women with various gynecological conditions requiring surgery, 1,880 were diagnosed with fibroids and 219 (12%) of them had endometriosis [27], while in a Thai study in which 331 women underwent surgery for benign gynecological diseases, a high proportion (28%) had co-existing endometriosis and uterine leiomyoma [28]. The proportion was much higher among 208 US women who underwent surgery for symptomatic leiomyoma between March 2011 and December 2015 at a single tertiary medical center, where 181 (87.1%) were found to have endometriosis [29]. All of these studies are limited by their small sample sizes or relatively short follow-up periods. The existing literature contains no large studies with long-term follow-up that have investigated the association between leiomyoma, endometriosis and other comorbidities, such as tube-ovarian infection, endometritis, infertility, autoimmune diseases, allergic diseases, breast cancer, cervical cancer, and ovarian cancer.

The purpose of our study was to investigate the co-existence of endometriosis and leiomyoma using long-term data from Taiwan's National Health Insurance Research Database (NHIRD). We specifically sought to determine whether having uterine leiomyoma increases the risk of developing endometriosis. We also examined the association between leiomyoma, endometriosis and other comorbidities in a large population from the same database.

## Materials and methods

### Data source

Taiwan's National Health Insurance (NHI) Program provides universal health insurance and covers almost all residents nationwide. Our data source comprised the Longitudinal Health Insurance Database 2000 (LHID2000), a representative subset of 1,000,000 randomly selected enrollees from the Registry for 23 million beneficiaries of the NHI program. The personal identification numbers in the claims data were recorded in electronic format before their release for research, to ensure that the privacy was protected for all the insured individuals. As the data used in this study were completely anonymized and de-identified before access and

analysis, there was no need to obtain informed patient consent. This study was approved by the Institutional Review Board of China Medical University Hospital in Taichung, Taiwan (CMUH104-REC2-115). Clinical diagnoses were classified based on the International Classification of Disease, Ninth Revision, Clinical Modification (ICD-9-CM) codes.

## Study participants

The case cohort consisted of women aged ≥20 years with a diagnosis of uterine leiomyoma (ICD-9-CM code 218) recorded in the LHID2000 between Jan 1, 2000 and Dec 31, 2012. The first recorded diagnosis of uterine leiomyoma served as the index date. Patients who had endometriosis before the index date or were aged less than 20 years were excluded. Controls were randomly selected from LHID2000 beneficiaries who did not have a recorded diagnosis of uterine leiomyoma and were frequency-matched (1:4) with cases within 5-year age groups and by the year of diagnosis. The detailed flowchart is presented in Fig 1.

The endpoint was a new diagnosis of endometriosis (ICD-9-CM code 617), which was defined as the first inpatient or outpatient date. The diagnosis of endometriosis was made by gynecologists when they identified endometriotic lesions during laparoscopy or ultrasound. All study participants were followed-up to the first occurrence of endometriosis, loss to follow-up, termination of NHI insurance, or until Dec 31, 2013, whichever was sooner.

Study participants were divided into three age groups: 20–39, 40–49, or ≥50 years. Several comorbidities were considered as potential confounders in this study, including tube-ovarian infection (ICD-9-CM code 614.2), endometritis (ICD-9-CM code 615), infertility (ICD-9-CM code 628), autoimmune diseases (ICD-9-CM code 710), allergic diseases (ICD-9-CM code 477, allergic rhinitis, which includes allergic rhinitis due to pollen [477.0], allergic rhinitis due to food [477.1], allergic rhinitis due to animal hair and dander [477.2], allergic rhinitis due to other allergen [477.8], and allergic rhinitis with cause unspecified [477.9]), breast cancer (ICD-9-CM code 174), cervical cancer (ICD-9-CM code 180), and ovarian cancer (ICD-9-CM code 183.0).

## Statistical analysis

The differences between uterine leiomyoma and control groups were examined by Chi-square testing for categorical variables and the Student's *t*-test for continuous variables. The Cox's proportional hazards model was used to estimate hazard ratios (HRs) and 95% confidence intervals (CIs) for assessing the risk of endometriosis-associated uterine leiomyoma. Multivariable models were adjusted for age and other comorbidities as mentioned above. The Kaplan-Meier method assessed the cumulative incidence of endometriosis in women with uterine leiomyoma and controls; the log-rank test determined the significance of difference between the two groups. A two-tailed p-value of <0.05 was considered statistically significant. We used SAS software (version 9.4 for Windows; SAS Institute Inc., Cary, NC, USA) for all data analyses.

## Results

We identified 31,239 women with uterine leiomyoma (cases) and 124,956 without leiomyoma (controls). Baseline variables for the study participants are shown in Table 1. Mean follow-up was 6.78±4.15 years for patients with uterine leiomyoma and 7.67±3.83 years for controls. Compared with controls, patients with uterine leiomyoma had significantly higher rates of endometritis, infertility, allergic diseases and breast cancer (p<0.001 for all comparisons). The incidence of cervical cancer was significantly lower in patients with leiomyoma than in controls (p<0.001).

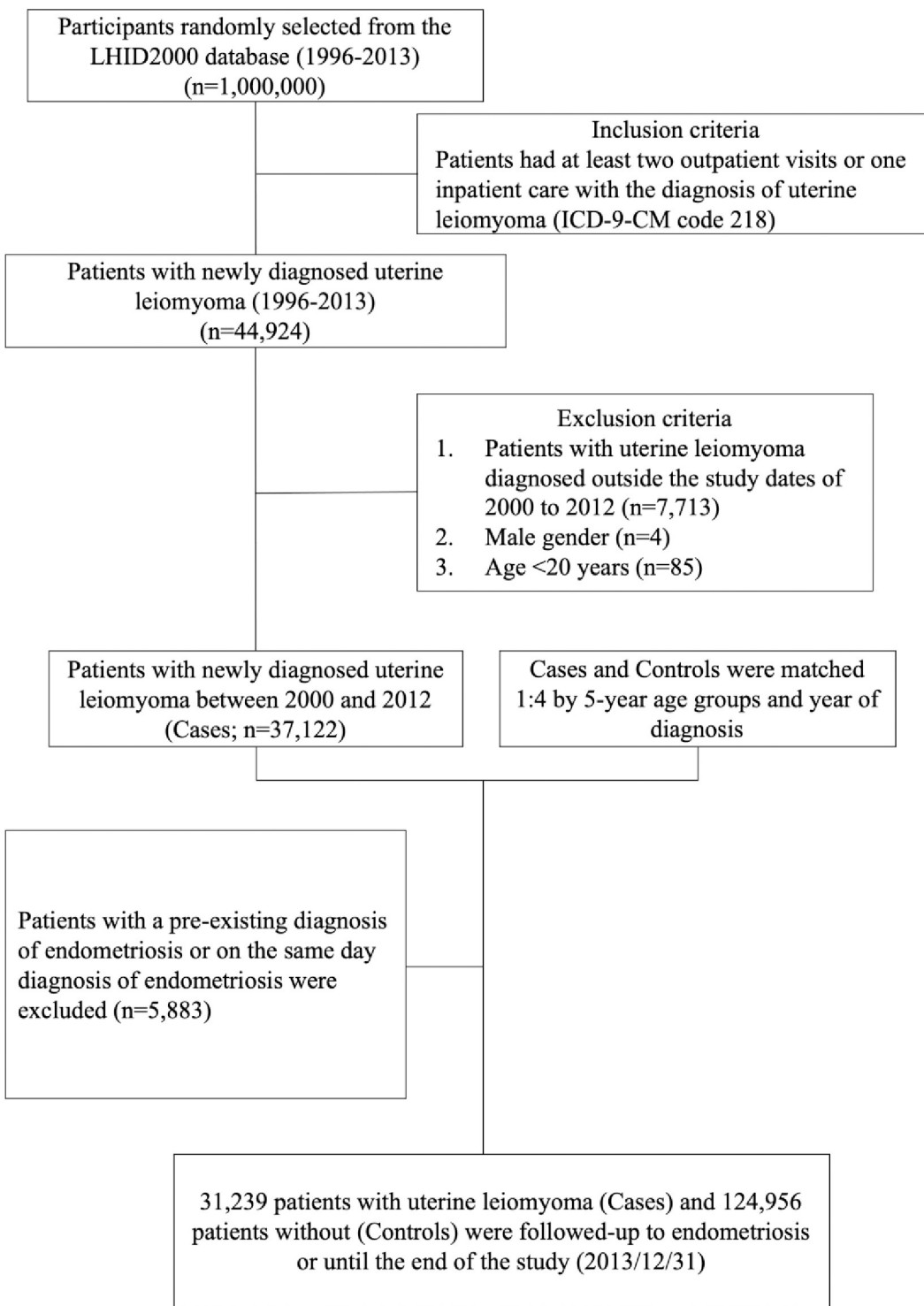

**Fig 1. Flowchart of study participants identified from the Longitudinal Health Insurance Database 2000 (LHID2000), a subset of Taiwan's National Health Insurance Research Database (NHIRD).**

In adjusted analyses, the risk of developing endometriosis was significantly higher among patients with uterine leiomyoma than controls (adjusted HR [aHR] 6.44; 95% CI, 6.18, 6.72). Comorbidities that were significantly associated with endometriosis included tube-ovarian

**Table 1. Baseline variables for patients with uterine leiomyoma and controls.**

| Variable | Uterine leiomyoma | | | | p-value |
|---|---|---|---|---|---|
| | No (n = 124,956) | | Yes (n = 31,239) | | |
| | N | % | n | % | |
| Mean age (years, ±SD)[a] | 41.9±9.10 | | 42.5±8.23 | | 0.77 |
| **Age (years)** | | | | | |
| 20–39 | 46,052 | 36.9 | 11,119 | 35.6 | |
| 40–49 | 60,692 | 48.6 | 15,567 | 49.8 | |
| ≥50 | 18,212 | 14.6 | 4,553 | 14.6 | |
| **Comorbidities** | | | | | |
| Tube-ovarian infection | 38 | 0.03 | 7 | 0.02 | 0.46 |
| Endometritis | 6,008 | 4.81 | 2,354 | 7.54 | <0.001 |
| Infertility | 4,803 | 3.84 | 1,729 | 5.53 | <0.001 |
| Autoimmune diseases | 312 | 0.25 | 80 | 0.26 | 0.84 |
| Allergic diseases | 22,258 | 17.8 | 6,735 | 21.6 | <0.001 |
| Cancer | 966 | 0.77 | 283 | 0.91 | 0.02 |
| Breast cancer | 670 | 0.54 | 266 | 0.85 | <0.001 |
| Cervical cancer | 271 | 0.22 | 13 | 0.04 | <0.001 |
| Ovarian cancer | 25 | 0.02 | 4 | 0.01 | 0.40 |
| Follow-up (years) mean±SD | 7.67±3.83 | | 6.78±4.15 | | <0.001 |

SD = standard deviation. Chi-square test

[a] Student's *t*-test.

infection (aHR 2.86; 95% CI, 1.28, 6.36), endometritis (1.14; 1.06, 1.24), infertility (1.26; 1.16, 1.37), and allergic diseases (1.11; 1.05, 1.17) (Table 2).

As shown in Table 3, the incidence rate of endometriosis in patients with uterine leiomyoma was 25.1 per 1,000 person-years, compared with 3.87 per 1,000 person-years in the control group. In fully adjusted analyses stratified by age and comorbidities, the risk of endometriosis was consistently significantly higher for those with uterine leiomyoma than for controls.

The data in Table 4 reveal a synergistic effect between uterine leiomyoma and comorbidities for the risk of endometriosis. Patients with both uterine leiomyoma and endometritis had a higher risk of developing endometriosis (aHR 6.73; 95% CI, 6.07, 7.45) than those with uterine leiomyoma alone (6.61; 6.33, 6.91) or endometritis alone (1.49; 1.31, 1.69). Similarly, patients with both uterine leiomyoma and infertility had a higher risk of endometriosis (aHR 6.95; 95% CI, 6.21, 7.78) than those with uterine leiomyoma only (6.66; 6.38, 6.96) or infertility only (1.78; 1.57, 2.02). No such synergistic effects were observed for allergic diseases and breast cancer; patients with uterine leiomyoma were apparently at significantly higher risk of developing endometriosis, whether or not they had either of those comorbidities.

As illustrated in Fig 2, Kaplan-Meier analysis revealed that the cumulative incidence of endometriosis was significantly higher in patients with uterine leiomyoma than in those without (log-rank test, p<0.001).

## Discussion

This large-scale nationwide study from Taiwan with 14 years of follow-up reveals that women at any age with uterine leiomyoma appear to be at significantly increased risk of developing endometriosis. Endometriosis and uterine fibroids are common disorders that affect

**Table 2. Crude and adjusted risks of developing endometriosis associated with uterine leiomyoma.**

| Variable | Endometriosis (n = 9,021) | Crude HR[†] | (95% CI) | p-value | Adjusted HR[††] | (95% CI) | p-value |
|---|---|---|---|---|---|---|---|
| **Uterine leiomyoma** | | | | | | | |
| No | 3,705 | 1.00 | Reference | | 1.00 | Reference | |
| Yes | 5,316 | 6.31 | (6.05, 6.58) | <0.001 | 6.44 | (6.18, 6.72) | <0.001 |
| **Age (years)** | | | | | | | |
| 20–39 | 4,268 | 1.00 | Reference | | 1.00 | Reference | |
| 40–49 | 4,409 | 4.99 | (4.47, 5.57) | <0.001 | 5.00 | (6.04, 6.57) | <0.001 |
| ≥50 | 344 | 3.75 | (3.36, 4.18) | <0.001 | 3.75 | (3.36, 4.18) | <0.001 |
| **Comorbidity** | | | | | | | |
| Tube-ovarian infection | | | | | | | |
| No | 9,015 | 1.00 | Reference | 0.02 | 1.00 | Reference | 0.01 |
| Yes | 6 | 2.60 | (1.17, 5.79) | | 2.86 | (1.28, 6.36) | |
| Endometritis | | | | | | | |
| No | 8,361 | 1.00 | Reference | <0.001 | 1.00 | Reference | <0.001 |
| Yes | 660 | 1.56 | (1.44, 1.69) | | 1.14 | (1.06, 1.24) | |
| Infertility | | | | | | | |
| No | 8,432 | 1.00 | Reference | <0.001 | 1.00 | Reference | <0.001 |
| Yes | 589 | 1.81 | (1.67, 1.97) | | 1.26 | (1.16, 1.37) | |
| Autoimmune diseases | | | | | | | |
| No | 9,001 | 1.00 | Reference | 0.80 | 1.00 | Reference | |
| Yes | 20 | 0.95 | (0.61, 1.47) | | | | |
| Allergic diseases | | | | | | | |
| No | 7,220 | 1.00 | Reference | <0.001 | 1.00 | Reference | <0.001 |
| Yes | 1801 | 1.27 | (1.20, 1.34) | | 1.11 | (1.05, 1.17) | |
| Breast cancer | | | | | | | |
| No | 8,993 | 1.00 | Reference | 0.003 | 1.00 | Reference | 0.02 |
| Yes | 28 | 0.58 | (0.40, 0.84) | | 0.64 | (0.44, 0.93) | |
| Cervical cancer | | | | | | | |
| No | 9,021 | 1.00 | Reference | | 1.00 | Reference | 1.00 |
| Yes | 0 | - | - | | - | - | - |
| Ovarian cancer | | | | | | | |
| No | 9,021 | 1.00 | Reference | | 1.00 | Reference | 1.00 |
| Yes | 0 | | | | | | |

HR = hazard ratio; CI = confidence interval.

[†]The crude HR represents the relative hazard ratio without adjustment for age and comorbidities.

[††]Variables found to be statistically significant in the univariable model were further examined in the multivariable model adjusted for age and comorbidities of tube-ovarian infection, endometritis, infertility, autoimmune diseases, allergic diseases, and breast cancer.

significant numbers of women. Previous studies have identified incidence rates of between 12% and 20% for concomitant fibroids and endometriosis [25, 30]. In a US study, 87% of 208 patients who underwent surgical treatment for symptomatic leiomyoma were found to have endometriosis [31]. Endometriosis poses a significantly high economic burden, both before and after diagnosis. A large longitudinal analysis of US healthcare claims databases observed that women with endometriosis experienced significantly higher annual healthcare costs compared with women without endometriosis, with an overall cost difference of $26,305 (in 2010 US dollars) over 10 years; $7,028 in the 5 years before diagnosis and $19,277 in the 5 years

**Table 3. Incidence rates and risk of endometriosis for patients with and without uterine leiomyoma, stratified by demographic variables and comorbidities.**

| Variables | Uterine leiomyoma | | | | | | Compared to patients without uterine leiomyoma | |
|---|---|---|---|---|---|---|---|---|
| | No | | | Yes | | | Crude HR[†] | Adjusted HR[††] |
| | (n = 124,956) | | | (n = 31,239) | | | | |
| | Event | Person-years | IR[†] | Event | Person-years | IR[†] | (95% CI) | (95% CI) |
| Total | 3,705 | 957,831 | 3.87 | 5,316 | 211,858 | 25.1 | 6.31 (6.05, 6.58)*** | 6.44 (6.18, 6.72)*** |
| **Age group** | | | | | | | | |
| 20–39 | 1,905 | 343,497 | 5.55 | 2,363 | 72,458 | 32.6 | 5.76 (5.42, 6.12)*** | 5.65 (5.32, 6.01)*** |
| 40–49 | 1,753 | 478,685 | 3.66 | 2,656 | 107,239 | 24.8 | 6.51 (6.13, 6.92)*** | 6.46 (6.08, 6.86)*** |
| ≥50 | 47 | 135,650 | 0.35 | 297 | 32,161 | 9.23 | 26.2 (19.3, 35.6)*** | 26.2 (19.2, 35.6)*** |
| **Comorbidities** | | | | | | | | |
| No | 2,600 | 764,399 | 3.40 | 3,697 | 154,990 | 23.9 | 6.85 (6.51, 7.20)*** | 7.13 (6.78, 7.50)*** |
| Yes | 1,105 | 193,433 | 5.71 | 1,619 | 56,867 | 28.5 | 4.94 (4.58, 5.33)*** | 5.09 (4.72, 5.50)*** |

IR = incidence rates per 1,000 person-years; HR = hazard ratio; CI = confidence interval.

[†]The crude HR represents the relative hazard ratio without adjustment for age and comorbidities.

[††]Variables found to be statistically significant in the univariable model were further examined in the multivariable model adjusted for age and comorbidities of tube-ovarian infection, endometritis, infertility, autoimmune diseases, allergic diseases, and breast cancer.

*$p < 0.05$; **$p < 0.01$

***$p < 0.001$.

after diagnosis [30]. It is important to evaluate the risk factors for endometriosis, to slow the inflammatory process and prevent the development of endometriosis. Early diagnosis and treatment of endometriosis might lessen the healthcare burden of this chronic disease.

To our knowledge, our research is the largest population-based longitudinal study with long-term follow-up to have examined the potential association between uterine leiomyoma and endometriosis. Our findings reveal that a strong association appears to exist between uterine leiomyoma and endometriosis. Our recent confirmation of an increased risk of endometriosis in patients with endometritis [32] led us to include endometritis as a potential confounder in this study. When we compared baseline disease variables between women with uterine leiomyoma and those without, we found significantly higher rates of endometritis, infertility and allergic diseases in the uterine leiomyoma cohort; in analyses that adjusted for all potential confounders, the risk of developing endometriosis was significantly higher in patients with uterine leiomyoma. Uterine leiomyoma was associated with a significantly increased risk of developing endometriosis in all age groups (20–39 years, 40–49 years, ≥50 years).

The reasons for the apparently strong association between leiomyoma and endometriosis are unclear. Researcher groups have identified risk factors for the coexistence of myomas and endometriosis, including younger age, nulliparity, moderate-to-severe pain, short menstrual intervals, smaller myomas, and locations of myomas [26, 33]. Our hypothesis is that uterine leiomyoma could predispose to retroverted uterus or distort the uterine cavity, increasing the risk of retrograde menstruation and thus increasing the risk of developing endometriosis. To support our theory, further studies are needed to investigate whether the numbers, sizes or locations of uterine leiomyoma impact differently on the risk of developing endometriosis.

It is well known that infertile women have a higher prevalence of endometriosis and are significantly more likely than fertile women to have moderate-to-severe endometriosis [34]. Similarly, our findings showed that women with infertility problems have an almost 2-fold greater risk of developing endometriosis, which increased to almost 7-fold in women with infertility and uterine leiomyoma. Having both comorbidities synergistically increased the risk of

**Table 4. Cox's proportional hazard regression analysis identified a synergistic effect between uterine leiomyoma and comorbidities for the risk of endometriosis.**

| Variables | | No. | Endometriosis | Adjusted HR[†] | p-value[#] |
|---|---|---|---|---|---|
| | | | No. | (95% CI) | |
| Uterine leiomyoma | Endometritis | | | | <0.001 |
| No | No | 118,948 | 3,454 | 1 (Reference) | |
| No | Yes | 6,008 | 251 | 1.49 (1.31, 1.69)*** | |
| Yes | No | 28,885 | 497 | 6.61 (6.33, 6.91)*** | |
| Yes | Yes | 2,354 | 409 | 6.73 (6.07, 7.45)*** | |
| Uterine leiomyoma | Infertility | | | | <0.001 |
| No | No | 120,153 | 3,451 | 1 (Reference) | |
| No | Yes | 4,803 | 254 | 1.78 (1.57, 2.02)*** | |
| Yes | No | 29,510 | 4,981 | 6.66 (6.38, 6.96)*** | |
| Yes | Yes | 1,729 | 335 | 6.95 (6.21, 7.78)*** | |
| Uterine leiomyoma | Allergic diseases | | | | <0.001 |
| No | No | 102,698 | 2,959 | 1 (Reference) | |
| No | Yes | 22,258 | 746 | 1.29 (1.19, 1.40)*** | |
| Yes | No | 24,504 | 4,261 | 6.79 (6.48, 7.12)*** | |
| Yes | Yes | 6,735 | 1,055 | 6.70 (6.24, 7.19)*** | |
| Uterine leiomyoma | Breast cancer | | | | 0.63 |
| No | No | 124,286 | 3,698 | 1 (Reference) | |
| No | Yes | 670 | 7 | 0.54 (0.26, 1.12) | |
| Yes | No | 30,973 | 5,295 | 6.44 (6.17, 6.72)*** | |
| Yes | Yes | 266 | 21 | 4.27 (2.78, 6.56)*** | |

HR = hazard ratio; CI = confidence interval.

[†]The Cox's model was adjusted for age and comorbidities of tube-ovarian infection, endometritis, infertility, autoimmune diseases, allergic diseases, and breast cancer.

*p<0.05; **p<0.01

***p<0.001.

developing endometriosis compared with women with infertility or uterine leiomyoma alone. Similarly, the risk for developing endometriosis was synergistically increased in women with uterine leiomyoma and endometritis. However, no such synergistic effect was observed among women with the comorbidities of breast cancer and allergic diseases.

This study has a number of strengths. It obtained data from a large number of participants from throughout Taiwan and provided long-term follow-up, allowing a high statistical power calculation (>0.9) for ascertaining an association between uterine leiomyoma and endometriosis. Stratified sensitivity analyses performed to clarify misclassifications and potential confounders did not reveal any significant changes in HRs between the different models. The potential association among uterine leiomyoma, endometriosis and other comorbidities has never previously been examined. Interestingly, our results demonstrated that a synergistic effect exists amongst these disorders.

There are some limitations with this study. The NHIRD dataset is not able to provide detailed demographic information in regard to laboratory results, levels of physical activity, or any family medical history, any of which may affect the apparent association observed between uterine leiomyoma and endometriosis. Moreover, the level of severity of endometriosis or uterine leiomyoma could not be analyzed from the NHIRD data. Another limitation is that in Taiwan, the oral contraceptive pill may be obtained over the counter, and it is not uncommon for women to purchase Chinese medicines, some of which may contain phytoestrogens with unknown clinical therapeutic effects. This sort of information is not provided by the NHIRD

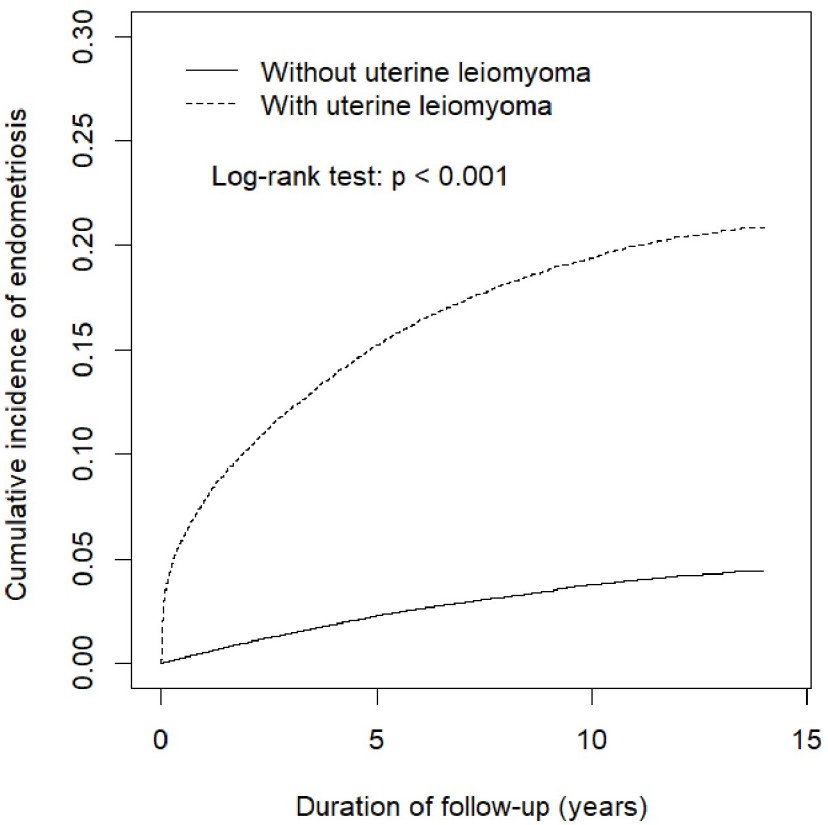

**Fig 2. Kaplan-Meier analysis of endometriosis risk in patients with and without uterine leiomyoma.** HR = hazard
ratio; CI = confidence interval.

database. However, our large sample size in this study may offset any population variation.
The apparent relationship between leiomyoma, endometriosis and hormonal therapy can be
further explored in future studies.

As for the practical implications in regard to the apparent association between uterine leio-
myoma and endometriosis, clinicians need to be aware of the symptoms that patients may
experience. Endometriosis should be suspected in women who have uterine leiomyoma with a
history of endometriosis or infertility. It is important for clinicians to have preoperative discus-
sions with a patient undergoing elective surgery for uterine leiomyoma, in order to gain prior
informed consent for removal of endometrial tissue discovered during surgery. Lastly, for an
infertile couple who is considering *in vitro* fertilization (IVF) or who has experienced unsuc-
cessful IVF attempts, ultrasound examinations should look for evidence of leiomyoma and
endometriosis.

## Conclusion

In conclusion, women at any age with uterine leiomyoma appear to be at significantly
increased risk of developing endometriosis. The presence of uterine leiomyoma with other
comorbidities, such as infertility or endometritis, synergistically increases the risk for endome-
triosis. Failure to diagnose and treat all etiologies of a patient's symptoms may lead to higher
treatment failure rates as well as an increased recurrence rate. Early recognition and treatment
of endometriosis among women with uterine leiomyoma may improve the clinical outcome,

slow the progression of endometriosis and alleviate the healthcare burden of this chronic disease.

## Acknowledgments

We thank all the patients and their guardians who participated in this research. We would also like to thank Iona J. MacDonald from China Medical University, Taichung, Taiwan, for her editing of this manuscript.

## Author Contributions

**Conceptualization:** Kent Yu-Hsien Lin, Wu-Chou Lin.

**Data curation:** Kent Yu-Hsien Lin, Chih-Yi Yang, Alan Lam, Cherry Yin-Yi Chang.

**Formal analysis:** Chih-Yi Yang, Cherry Yin-Yi Chang.

**Investigation:** Cherry Yin-Yi Chang, Wu-Chou Lin.

**Methodology:** Chih-Yi Yang.

**Supervision:** Alan Lam.

**Writing – original draft:** Kent Yu-Hsien Lin, Chih-Yi Yang.

**Writing – review & editing:** Cherry Yin-Yi Chang, Wu-Chou Lin.

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
