## [Decision Letter · Decision Letter 0]

4 May 2021

PONE-D-21-09322

Uterine leiomyoma is associated with the risk of developing endometriosis: a nationwide cohort study involving 156,195 women

PLOS ONE

Dear Dr. Yang,

Thank you for submitting your manuscript to PLOS ONE. After careful consideration, we feel that it has merit but does not fully meet PLOS ONE’s publication criteria as it currently stands. Therefore, we invite you to submit a revised version of the manuscript that addresses the points raised during the review process.

We look forward to receiving your revised manuscript.

Kind regards,

Antonio Simone Laganà, M.D., Ph.D.

Academic Editor

PLOS ONE

Journal Requirements:

Thank you for stating the following in the Funding Section of your manuscript:

This study was supported in part by Taiwan’s Ministry of Health and Welfare Clinical Trial Center (MOHW109-TDU-B-212-114004), the MOST Clinical Trial Consortium for Stroke (MOST 108-2321-B-039-003-) and the Tseng-Lien Lin Foundation, Taichung, Taiwan.

Please include captions for your Supporting Information files at the end of your manuscript, and update any in-text citations to match accordingly. Please see our Supporting Information guidelines for more information: http://journals.plos.org/plosone/s/supporting-information.

Additional Editor Comments:

The topic of the manuscript is interesting. Nevertheless, the reviewers raised several concerns: considering this point, I invite authors to perform the required major revisions.

Reviewers' comments:

Reviewer's Responses to Questions

**Comments to the Author**

1. Is the manuscript technically sound, and do the data support the conclusions?

Reviewer #1: Yes

Reviewer #2: Yes

Reviewer #3: No

2. Has the statistical analysis been performed appropriately and rigorously? 

Reviewer #1: Yes

Reviewer #2: Yes

Reviewer #3: No

3. Have the authors made all data underlying the findings in their manuscript fully available?

Reviewer #1: Yes

Reviewer #2: Yes

Reviewer #3: Yes

4. Is the manuscript presented in an intelligible fashion and written in standard English?

Reviewer #1: Yes

Reviewer #2: Yes

Reviewer #3: Yes

5. Review Comments to the Author

Reviewer #1: PONE-D-21-09322

The manuscript entitled “Uterine leiomyoma is associated with the risk of developing endometriosis: a nationwide cohort study involving 156,195 women” analyzed the association between uterine leiomyoma and the increased risk for endometriosis.

The study is well written and the material and methods are correctly used. I suggest few corrections as detailed below:

In the Introduction, Authors should underline the role of quality of life in endometriosis, briefly referring to PMID: 31755667, PMID: 30269659, and PMID: 31726815

The first sentence of the Discussion should explain the main findings of the study.

Reviewer #2: I was pleased to review the Research Article “Uterine leiomyoma in associated with the risk of developing endometriosis: a nationwide cohort study involving 156, 195 women.” from PLOS ONE.

The methodology used by the Authors is appropriate for the purpose of the study and conclusions are narrow linked to data discussion and available evidence. The English language is fluid and well understood. Nevertheless, major revision is needed, but the paper is of good quality.

I suggest this:

1. Abstract: is complete and well written.

2. Introduction: I really appreciated the quote over the theory about retrograde menstruation, and it is important to emphasize the association, as you have done, concerning estrogenic dysregulation: uterine leiomyoma and endometriosis are hormone-dependent. I suggest taking into account these papers to improve the quality of this section: doi: 10.3390/ijerph17134683; doi: 10.1093/biolre/ioab054.

3. Material and Methods. the strong point of the work structure is the big data source comprised the Longitudinal Health Insurance Database 2000 and the study participants with women aged > 20 years between 2000 and 2012.

4. Results. I have some questions.

• Were patients with endometritis treated? if yes, how?

• When you talk about the association between leiomyoma and infertility which increases the risk of endometriosis, what about the patients who had infertility problems in detail?

• When you define the non-synergy on the risks associated with allergies, what kind of allergies are we talking about?

• When you talk about non-synergy with breast cancer, did the patients have BRCA mutations?

5. Discussion. Good ideas for future work, as this leiomyoma-endometriosis association is clear…but why…this exists, not yet.

Also as you wrote, the database does not contain: demographic details, levels of physical activity, or any family medical history, another weak point as you said, is the level of severity of endometriosis or uterine leiomyoma could not be analyzed, another point is the not uncommon use for these women of the oral contraceptive pill.

6. Conclusion. I found the conclusions coherent with the work done. Excellent reflection for an infertile couple who is considering IVF to investigate the presence of leiomyoma/endometriosis.

Reviewer #3: I was pleased to revise the manuscript entitled “Uterine leiomyoma is associated with the risk of developing endometriosis: a nationwide cohort study involving 156,195 women” (Manuscript Number: PONE-D-21-09322).

Taiwan’s Ministry of Health and Welfare Clinical Trial Center (MOHW109-TDU-B-212-114004), the MOST Clinical Trial Consortium for Stroke (MOST 108-2321-B-039-003-) and the Tseng-Lien Lin Foundation, Taichung, Taiwan approved the study.

In my honest opinion, the topic is interesting enough to attract the readers’ attention. Nevertheless, the manuscript may benefit from major revisions:

- The text needs a language revision to improve the readability, some typos, and some grammatical errors.

- Abstract. The retrospective study design cannot support the conclusion that the risk of endometriosis is increased by uterine leiomyoma. Based on the study methods, it is correct to refer only to the “association” between a diagnosis of leiomyoma and endometriosis.

- I would suggest checking the use of abbreviation that should be reported in the extended form at the first use in the abstract, text, and tables.

- Please, clarify which type of diagnosis of myoma were included. Any modality?

- The used study methods cannot provide an answer to the question of whether having uterine leiomyoma increases the risk of developing endometriosis. Only an association is observable. A cause-effect relationship is not possible. Please, revise the used terms.

- Lines 131-134. Please, clarify what you mean by “who did not have a recorded diagnosis of uterine leiomyoma”. If you refer that matched cohort did not diagnose uterine myomas for the entire study period, this is wrong. Matched cohort should include women without a diagnosis of myoma up to the index date; otherwise, you introduce a bias in the study design if myomas cannot be diagnosed later. Indeed, an appropriate study design can include patients both as exposed and unexposed. One patient should be matched with women that the index year who had the same age and not a diagnosis of myoma. These women can become exposed cohort when 5 years later have a diagnosis of myoma and will be matched with women that the index date (year of myoma diagnosis) have the same age and never had a diagnosis of myoma. Differently, in the matched cohort, you are artificially selecting women who will never develop myoma, which is not natural considering the high proportion of women who in life develop uterine fibroids.

- Lines 180-188. I would suggest better clarifying these study results. The association regarding time relationship.

- Please, report HR and 95%CI in all cases.

- Abstract and lines 263-267. This statement is unclear because in the present study, cases with a diagnosis of endometriosis concomitant to the diagnosis of myoma were excluded (figure 1).

- I would suggest discussing better, at least briefly, the therapeutic options for endometriosis and its pathogenesis. Refer to PMID: 32046116; PMID: 31532753; PMID: 33096011; PMID: 32981374

6. PLOS authors have the option to publish the peer review history of their article (what does this mean?). If published, this will include your full peer review and any attached files.

Reviewer #1: No

Reviewer #2: No

Reviewer #3: No

---

## [Author Response · Author response to Decision Letter 0]

9 Jul 2021

Reference: PONE-D-21-09322

Academic Editor, PLOS One

Dear Dr. Laganà

We greatly appreciate the comments from your Reviewers on our manuscript Uterine leiomyoma is associated with the risk of developing endometriosis: a nationwide cohort study involving 156,195 women (Ref.: PONE-D-21-09322). 

We have carefully revised the manuscript according to the suggestions raised by the Reviewers; specific points are addressed below. We have marked up our changes using red font in the revised manuscript.

We sincerely hope that our revised manuscript is now suitable for publication in PLOS One.

Yours sincerely,

Kent Yu-Hsien Lin (M.B.B.S., M.Med) and Chih-Yi Yang (M.D.)

Reviewer #1: PONE-D-21-09322

The manuscript entitled Uterine leiomyoma is associated with the risk of developing endometriosis: a nationwide cohort study involving 156,195 women analyzed the association between uterine leiomyoma and the increased risk for endometriosis.

The study is well written and the material and methods are correctly used. I suggest few corrections as detailed below:

In the Introduction, Authors should underline the role of quality of life in endometriosis, briefly referring to PMID: 31755667, PMID: 30269659, and PMID: 31726815

The first sentence of the Discussion should explain the main findings of the study.

Response: We thank the Reviewer for this constructive feedback, which has enabled us to improve our manuscript. We have accordingly amended the Introduction and Discussion, as follows:

These symptoms can have enormous impact on a sufferer’s quality of life, with well-documented evidence describing how these symptoms interfere with daily life and activities, adversely affecting general health and psychological well-being, and impacting on overall sexual functioning (13-15). (Lines 77-80)

This large-scale nationwide study from Taiwan with 14 years of follow-up reveals that women at any age with uterine leiomyoma appear to be at significantly increased risk of developing endometriosis. (Lines 228-230)

Reviewer #2: I was pleased to review the Research Article Uterine leiomyoma in associated with the risk of developing endometriosis: a nationwide cohort study involving 156, 195 women. from PLOS ONE.

The methodology used by the Authors is appropriate for the purpose of the study and conclusions are narrow linked to data discussion and available evidence. The English language is fluid and well understood. Nevertheless, major revision is needed, but the paper is of good quality.

Response: We thank the Reviewer for this thoughtful, appreciative feedback on our manuscript.

Reviewer #2: I suggest this:

1. Abstract: is complete and well written.

Response: We thank the Reviewer for this approval of the Abstract. 

2. Introduction: I really appreciated the quote over the theory about retrograde menstruation, and it is important to emphasize the association, as you have done, concerning estrogenic dysregulation: uterine leiomyoma and endometriosis are hormone-dependent. I suggest taking into account these papers to improve the quality of this section: doi: 10.3390/ijerph17134683; doi: 10.1093/biolre/ioab054.

Response: We thank the Reviewer for these insightful suggestions and we have accordingly amended our text, adding more references in regard to the adverse impact of endometriosis-related symptoms on quality of life, as this was called for by another Reviewer as well. Our amended section in the Introductory text is as follows:

Endometriosis is another common gynecological condition, affecting 10–50% of reproductive-aged women (7-9). The most common symptoms are dysmenorrhea, chronic pelvic pain, dyspareunia and subfertility (10-12). These symptoms can have enormous impact on a sufferer’s quality of life, with well-documented evidence describing how these symptoms interfere with daily life and activities, adversely affecting general health and psychological well-being, and impacting on overall sexual functioning (13-15). Moreover, endometriosis-related symptoms impose a substantial economic burden on those afflicted in terms of direct costs of medical care and indirect costs, such as costs for sick leave and work absenteeism (14, 16). It is well known that the risk of endometriosis is higher in young women with menstrual dysfunction and any obstruction to the outflow of menstrual products (17, 18). This is thought to be because of the higher volume of refluxed menstrual blood and endometrial tissue fragments, lending support to the theory that endometriosis results from retrograde menstruation (17). Genetic insights, the possible roles of the environment and the immune system, as well as intrinsic abnormalities in the endometrium of affected women have helped to elucidate the pathogenesis of endometriosis (19). Endometriosis is an estrogen-dependent disorder. Recent speculations suggest that the estrogen-dependent activity of different ion channels may contribute to the etiology of endometriosis (20). The levels of estrogen expression and functionality of these ion channels fluctuate between menstrual phases, with increased estrogen enhancing pathogenic aspects of endometriosis, such as the proliferation and in vivo implantation of ectopic endometrium (20). More evidence is needed to determine the strength of this association. Available conventional therapeutic strategies for managing endometriosis (surgery, hormonal treatment, nonsteroidal anti-inflammatories) have limited or intermittent efficacy, as well as unwanted hormonal side effects such as the loss of bone density (21). Aromatase inhibitors (AIs) have been proposed for endometriosis treatment, with limited clinical data indicating associated reductions in volume and symptoms of endometrioma, but AIs have also been linked to the development of functional cysts and menopausal symptoms, as well as reductions in bone mineral density (22). Alternative progestins and estroprogestins drug delivery methods (e.g., vaginal rings, patches, subcutaneous implants, intrauterine drug delivery systems, nanotechnologies) offer potential advantages of fewer adverse events, greater therapeutic efficacy, improved patient compliance and satisfaction (23). At this point, more clinical experience is needed before any conclusions can be made about the clinical efficacies of these alternative drug delivery methods (23). Thus, the disadvantages of current pharmacotherapeutic options have induced many women with endometriosis to seek alternative treatment options, such as phytotherapy (21, 24). Promisingly, in vitro and preclinical investigations have indicated that some phytochemicals demonstrate strong phytoestrogenic effects capable of modulating estrogen and inflammatory activity, which could help to reduce endometriosis symptoms, although randomized controlled trial data are lacking (21, 24). Interestingly, increasing clinical and pathophysiological evidence supports the notion that chronic inflammation is associated with benign gynecological disorders such as uterine leiomyoma and endometriosis (25). These associations have been comprehensively reviewed (25), although basic and clinical investigations have yet to clearly define the mechanism responsible for the development of endometriosis. Up until now, research has focused on altered immune responses and other regulatory factors that may mediate the histogenesis of endometriosis. (Lines 77-118)

3. Material and Methods. the strong point of the work structure is the big data source comprised the Longitudinal Health Insurance Database 2000 and the study participants with women aged > 20 years between 2000 and 2012.

Response: We thank the Reviewer for highlighting the fact that the body of evidence from the LHID2000 lends weight to our study. 

4. Results. I have some questions.

(i) Were patients with endometritis treated? if yes, how?

(ii) When you talk about the association between leiomyoma and infertility which increases the risk of endometriosis, what about the patients who had infertility problems in detail?

(iii) When you define the non-synergy on the risks associated with allergies, what kind of allergies are we talking about?

(iv) When you talk about non-synergy with breast cancer, did the patients have BRCA mutations?

Responses: 

(i) Our endometritis data were collected according to coding by the International Classification of Disease, Ninth Revision, Clinical Modification (ICD-9-CM code 615), classifying the disease by the patient’s signs and symptoms. Indeed, all patients were treated with antibiotics.

(ii) Our infertility data were collected according to coding by the International Classification of Disease, Ninth Revision, Clinical Modification (ICD-9-CM code 628), classifying the disease by the patient’s history and symptoms. Unfortunately, we are unable to collect the different causes of infertility in our database. Our Discussion section mentions that in patients with infertility problems, ultrasound examinations should look for evidence of leiomyoma and endometriosis. (Lines 302-305)

(iii) For allergic diseases, we used ICD-9-CM code 477, allergic rhinitis, which includes allergic rhinitis due to pollen (477.0), allergic rhinitis due to food (477.1), allergic rhinitis due to animal hair and dander (477.2), allergic rhinitis due to other allergen (477.8), and allergic rhinitis with cause unspecified (477.9). We have now added this information to the manuscript, to clarify for the readers what we mean by “allergic diseases”. (Lines 177-179)

(iv) According to ICD-9-CM coding, we included all types of breast cancers in our study, which include but are not limited to estrogen receptor-positive, progesterone receptor-positive, hormone receptor-negative, HER2-positive, BRCA mutations, and so on. 

5. Discussion. Good ideas for future work, as this leiomyoma-endometriosis association is clear &but why &this exists, not yet.

Also as you wrote, the database does not contain: demographic details, levels of physical activity, or any family medical history, another weak point as you said, is the level of severity of endometriosis or uterine leiomyoma could not be analyzed, another point is the not uncommon use for these women of the oral contraceptive pill.

Response: We greatly appreciate the Reviewer’s close attention to our work. 

6. Conclusion. I found the conclusions coherent with the work done. Excellent reflection for an infertile couple who is considering IVF to investigate the presence of leiomyoma/endometriosis.

Response: We sincerely thank the Reviewer for this endorsement of our concluding recommendations. 

Reviewer #3: I was pleased to revise the manuscript entitled Uterine leiomyoma is associated with the risk of developing endometriosis: a nationwide cohort study involving 156,195 women (Manuscript Number: PONE-D-21-09322).

Taiwan’s Ministry of Health and Welfare Clinical Trial Center (MOHW109-TDU-B-212-114004), the MOST Clinical Trial Consortium for Stroke (MOST 108-2321-B-039-003-) and the Tseng-Lien Lin Foundation, Taichung, Taiwan approved the study.

In my honest opinion, the topic is interesting enough to attract the readers attention. 

Response: We thank the Reviewer for this endorsement of our work. 

Reviewer #3: Nevertheless, the manuscript may benefit from major revisions:

- The text needs a language revision to improve the readability, some typos, and some grammatical errors.

Response: The manuscript has been edited by a native English speaker who also has several years of medical editing and writing experience.

- Abstract. The retrospective study design cannot support the conclusion that the risk of endometriosis is increased by uterine leiomyoma. Based on the study methods, it is correct to refer only to the association between a diagnosis of leiomyoma and endometriosis.

Response: We thank the Reviewer for making this point and we have accordingly amended our Abstract conclusion text as follows:

A diagnosis of uterine leiomyoma appears to increase the risk of endometriosis. (Lines 64-65)

- I would suggest checking the use of abbreviation that should be reported in the extended form at the first use in the abstract, text, and tables.

Response: Thank you for paying such close attention to our text. We have accordingly checked all of the text and amended it wherever necessary where an abbreviation needed to be defined at first mention. 

- Please, clarify which type of diagnosis of myoma were included. Any modality?

Response: Four types of myoma were included: submucosal; intramural; subserosal; and pedunculated. Any type of myoma was included in our study and all were diagnosed by ultrasound or surgical methods.

- The used study methods cannot provide an answer to the question of whether having uterine leiomyoma increases the risk of developing endometriosis. Only an association is observable. A cause-effect relationship is not possible. Please, revise the used terms.

Response: We thank the Reviewer for pointing this out and we have accordingly revised the conclusions text in the Abstract, text in the Results section, and text in the Discussion. (Lines 64-65, 221, 229, 245-246, 257, 281, 294, 297, 308 and 313) 

- Lines 131-134. Please, clarify what you mean by who did not have a recorded diagnosis of uterine leiomyoma. If you refer that matched cohort did not diagnose uterine myomas for the entire study period, this is wrong. Matched cohort should include women without a diagnosis of myoma up to the index date; otherwise, you introduce a bias in the study design if myomas cannot be diagnosed later. Indeed, an appropriate study design can include patients both as exposed and unexposed. One patient should be matched with women that the index year who had the same age and not a diagnosis of myoma. These women can become exposed cohort when 5 years later have a diagnosis of myoma and will be matched with women that the index date (year of myoma diagnosis) have the same age and never had a diagnosis of myoma. Differently, in the matched cohort, you are artificially selecting women who will never develop myoma, which is not natural considering the high proportion of women who in life develop uterine fibroids.

Response: Our matched cohort (control group) only included women without a diagnosis of myoma up to the index date, which means that for every individual woman with no history of myoma up to the index date, we commenced her follow-up and continued until the end of 2012, to determine whether she developed endometriosis during the follow-up period. We followed-up 124,956 women in our control group. 

- Lines 180-188. I would suggest better clarifying these study results. The association regarding time relationship.

Response: To explain clearly how we calculated mean follow-up years, we can use the study group (leiomyoma group) as an example:

If a woman who had no pre-existing endometriosis disease was diagnosed with uterine leiomyoma in 2001, we followed-up this particular woman for years and realized that she developed endometriosis in 2011. Her follow-up time would thus be 11 years. On the other hand, if a woman without pre-existing endometriosis was diagnosed with leiomyoma in 2009 and we were only able to follow her until the end of 2012, her follow-up time would be 4 years. The average of the follow-up times in these two women is 7.5 years (11 years + 4 years = 15 years, divided by 2 = 7.5 years). The mean follow-up in these two women as our study group would thus be 7.5 years. Instead of using just two women, our study used follow-up times from 31,239 women to calculate the mean, arriving at 6.78 +/- 4.15 years in our study group (leiomyoma group). For the control group, as per the above, by matching the age and the year of diagnosis, we used 124,956 women to calculate the mean follow-up time, arriving at 7.67 +/- 3.83 years. 

- Please, report HR and 95%CI in all cases.

Response: We have accordingly enriched the Abstract text with adjusted HR and 95% CI values, which was the only place in the manuscript that had not reported all such values. Table 1 cannot provide HR/95% CI values, but does report p-values, which are essentially the same as 95% CIs. 

- Abstract and lines 263-267. This statement is unclear because in the present study, cases with a diagnosis of endometriosis concomitant to the diagnosis of myoma were excluded (figure 1).

Response: The aim of this research was to determine whether the presence of leiomyoma would increase the risk of developing endometriosis. Figure 1 includes 1 million patients randomly selected from our database. Among those 1 million patients, 44,924 patients had ‘newly’ diagnosed uterine leiomyoma. A total of 5,883 patients with a pre-existing diagnosis of endometriosis were excluded. Along with our exclusion criteria (7,713 + 4 + 85 = 7,802 patients), this yielded only 31,239 patients with uterine leiomyoma to conduct the follow-up study (44,924 – 7,802 – 5,883 = 31,239 patients). The follow-up study revealed for us the information that leiomyoma increases the risk of developing endometriosis. Hence, we suggested that women with infertility should look for underlying risk factors such as endometriosis and leiomyoma (as women with leiomyoma were likely to have endometriosis, according to our study evidence).

- I would suggest discussing better, at least briefly, the therapeutic options for endometriosis and its pathogenesis. Refer to PMID: 32046116; PMID: 31532753; PMID: 33096011; PMID: 32981374

Response: We thank the Reviewer for this suggestion, which enables us to improve our text. We have accordingly amended the Introduction to read as follows:

Genetic insights, the possible roles of the environment and the immune system, as well as intrinsic abnormalities in the endometrium of affected women have helped to elucidate the pathogenesis of endometriosis (19). Endometriosis is an estrogen-dependent disorder. Recent speculations suggest that the estrogen-dependent activity of different ion channels may contribute to the etiology of endometriosis (20). The levels of estrogen expression and functionality of these ion channels fluctuate between menstrual phases, with increased estrogen enhancing pathogenic aspects of endometriosis, such as the proliferation and in vivo implantation of ectopic endometrium (20). More evidence is needed to determine the strength of this association. Available conventional therapeutic strategies for managing endometriosis (surgery, hormonal treatment, nonsteroidal anti-inflammatories) have limited or intermittent efficacy, as well as unwanted hormonal side effects such as the loss of bone density (21). Aromatase inhibitors (AIs) have been proposed for endometriosis treatment, with limited clinical data indicating associated reductions in volume and symptoms of endometrioma, but AIs have also been linked to the development of functional cysts and menopausal symptoms, as well as reductions in bone mineral density (22). Alternative progestins and estroprogestins drug delivery methods (e.g., vaginal rings, patches, subcutaneous implants, intrauterine drug delivery systems, nanotechnologies) offer potential advantages of fewer adverse events, greater therapeutic efficacy, improved patient compliance and satisfaction (23). At this point, more clinical experience is needed before any conclusions can be made about the clinical efficacies of these alternative drug delivery methods (23). Thus, the disadvantages of current pharmacotherapeutic options have induced many women with endometriosis to seek alternative treatment options, such as phytotherapy (21, 24). Promisingly, in vitro and preclinical investigations have indicated that some phytochemicals demonstrate strong phytoestrogenic effects capable of modulating estrogen and inflammatory activity, which could help to reduce endometriosis symptoms, although randomized controlled trial data are lacking (21, 24). Interestingly, increasing clinical and pathophysiological evidence supports the notion that chronic inflammation is associated with benign gynecological disorders such as uterine leiomyoma and endometriosis (25). These associations have been comprehensively reviewed (25), although basic and clinical investigations have yet to clearly define the mechanism responsible for the development of endometriosis. (Lines 86-116)

---

## [Decision Letter · Decision Letter 1]

16 Aug 2021

Uterine leiomyoma is associated with the risk of developing endometriosis: a nationwide cohort study involving 156,195 women

PONE-D-21-09322R1

Dear Dr. Yang,

We’re pleased to inform you that your manuscript has been judged scientifically suitable for publication and will be formally accepted for publication once it meets all outstanding technical requirements.

Kind regards,

Antonio Simone Laganà, M.D., Ph.D.

Academic Editor

PLOS ONE

Additional Editor Comments (optional):

Authors performed the required corrections, which were positively evaluated by the reviewers. I am pleased to accept this paper for publication.

Reviewers' comments:

Reviewer's Responses to Questions

**Comments to the Author**

1. If the authors have adequately addressed your comments raised in a previous round of review and you feel that this manuscript is now acceptable for publication, you may indicate that here to bypass the “Comments to the Author” section, enter your conflict of interest statement in the “Confidential to Editor” section, and submit your "Accept" recommendation.

Reviewer #1: All comments have been addressed

Reviewer #2: All comments have been addressed

2. Is the manuscript technically sound, and do the data support the conclusions?

Reviewer #1: (No Response)

Reviewer #2: Yes

3. Has the statistical analysis been performed appropriately and rigorously? 

Reviewer #1: No

Reviewer #2: Yes

4. Have the authors made all data underlying the findings in their manuscript fully available?

Reviewer #1: Yes

Reviewer #2: Yes

5. Is the manuscript presented in an intelligible fashion and written in standard English?

Reviewer #1: Yes

Reviewer #2: Yes

6. Review Comments to the Author

Reviewer #1: The Authors improved the manuscript as requested. I suggest the acceptance of the manuscript in the present form.

Kind regards

Reviewer #2: Good job, all changes have been performed as requested. I only ask you to check the reference list in order to avoid duplicate

7. PLOS authors have the option to publish the peer review history of their article (what does this mean?). If published, this will include your full peer review and any attached files.

Reviewer #1: No

Reviewer #2: No

---

## [Editor Report · Acceptance letter]

18 Aug 2021

PONE-D-21-09322R1 

Uterine leiomyoma is associated with the risk of developing endometriosis: a nationwide cohort study involving 156,195 women 

Dear Dr. Yang:

I'm pleased to inform you that your manuscript has been deemed suitable for publication in PLOS ONE. Congratulations! Your manuscript is now with our production department. 

Kind regards, 

on behalf of

Dr. Antonio Simone Laganà 

Academic Editor

PLOS ONE